# Minimum dataset with integrated scoring and indexing methods for soil quality assessment

Khandakar Islam[1]*, Arifur Rahman[1], Warren Dick[2], Vinayak Shedekar[3],
Javier Gonzalez[4], Dexter Watts[5], Norman Fausey[6], Marvin Batte[7], Tara VanToai[6],
Randall Reeder[3], Dennis Flanagan[4]

**1** Soil, Water, and Bioenergy Resources, Ohio State University South Centers, Piketon, Ohio, United
States of America, **2** School of Environment and Natural Resources, Ohio State University, Wooster,
Ohio, United States of America, **3** Food, Agricultural and Biological Engineering, Ohio State University,
Columbus, Ohio, United States of America, **4** National Soil Erosion Research Laboratory, USDA-ARS,
West Lafayette, Indiana, United States of America, **5** National Soil Dynamics Laboratory, USDA-ARS,
Auburn, Alabama, United States of America, **6** Soil Drainage Research Unit, USDA-ARS, Columbus,
Ohio, United States of America, **7** Agricultural, Environmental and Developmental Economics, Ohio State
University, Columbus, Ohio, United States of America

* islam.27@osu.edu

journal.pone.0346136

Research Council, NEPAL

**Peer Review History:** PLOS recognizes the
benefits of transparency in the peer review
process; therefore, we enable the publication
of all of the content of peer review and
author responses alongside final, published
articles. The editorial history of this article is
available here: https://doi.org/10.1371/journal.
pone.0346136

## Abstract

Soil quality (SQ) is a key determinant of agricultural productivity and environmental
sustainability, yet its assessment is challenged by the diverse functions of soil and
the absence of universally accepted indicators. This study aimed to develop a crop
yield-correlated minimum dataset (MDS$_{Corr}$) for SQ assessment and evaluate its
performance across multiple U.S. regions. Over a five-year period, data (n = 576)
from geo-referenced composite soils at 0–30 cm depth were collected from gypsum
amended cover crop integrated corn-soybean rotation experimental sites at Shorter
(Alabama), Farmland (Indiana), and Hoytville and Piketon (Ohio). Using the available
soil and crop yield data, six scoring functions (four linear and two nonlinear) and
three indexing approaches (additive, weighted additive, and Nemoro) were evalu-
ated to calculate the SQ index (SQI). The MDS$_{Corr}$ identified a reduced set of key soil
properties most strongly associated with corn productivity, including total organic car-
bon, microbial biomass carbon, active carbon, total nitrogen, and aggregate-related
physical indicators explaining SQ. Using different scoring and indexing approaches,
the calculated SQI values at the Indiana site, used as a reference ranged from
0.31 to 0.6. Among the approaches, linear scoring with threshold limits and additive
indexing produced the most consistent SQI values, reducing variability to within ±1%
compared to the total dataset (TDS). The MDS$_{Corr}$-based SQI showed strong positive
correlations with the TDS-derived SQI (R² = 0.53 to 0.93) and outperformed the princi-
pal component analysis-based MDS (MDS$_{PCA}$) in terms of reliability and consistency.
Based on MDS$_{Corr}$-derived SQI values, the relative SQ rankings for the four study
sites were: Hoytville > Indiana > Alabama > Piketon. While calibration and validation
are recommended across geographic regions and cropping systems, the MDS$_{Corr}$

**Data availability statement:** All relevant data are provided in "Supporing Information" zip file.

**Funding:** United Soybean Board, Grant/Award Number: 1520-732-7226; The Ohio State University; USDA-ARS.

**Competing interests:** N/A.

approach, when combined with linear scoring and additive indexing, has the potential to provide a simplified and transferable framework for SQ assessment.

## Introduction

Soil quality (SQ) is an integration of inherent and dynamic capacities of soil to perform ecosystem functions such as supporting plant productivity, regulating environmental processes, and sustaining the health of plants and animals within natural or managed ecosystem boundaries [1–3]. It is a complex, multifaceted functional concept that cannot be measured directly but can be inferred from soil functions represented by core biological, chemical, and physical indicators [2,4]. These indicators represent soil properties and/or processes that are sensitive to management-induced changes in soil functions and associated ecosystem services [5–11]. However, due to the time, cost, equipment, labor, complexity, and interrelations involved in analyzing all soil properties, it is often impractical to include them all in SQ assessments. Consequently, selecting an appropriate minimum dataset (MDS) of key soil indicators is essential for effective and meaningful SQ evaluation [5,12].

Until now, no universally accepted set of soil indicators has been established for SQ assessment and monitoring, although attempts have been made to select and evaluate such indicators worldwide [1,13,14]. These efforts often yield highly variable and inconclusive estimates of SQ primarily due to wide variability of soil types and functions. Several studies have also employed a larger or total dataset (TDS) in combination with various scoring functions to create a SQ indexing (SQI) method. However, SQI values have shown considerable variability, even for the same type of soil, sampling locations and conditions, depending on the scoring and indexing approaches used [6,15–17]. Recent studies conducted under different cropping systems have reported improved sensitivity and effectiveness of nonlinear scoring functions combined with weighted additive indexing for soil quality assessment. For example, Iheshiulo et al. (2024) [18] demonstrated that a nonlinear weighted additive indexing approach outperformed alternative methods in capturing soil health responses under short-term crop rotations in the Canadian prairies. Due to the absence of existing standardized methodologies to assess SQ, and to simplify and reduce the labor and cost required to assess SQ, a minimum dataset (MDS) approach is suggested as a practical way to make SQ assessments [19,20].

Studies relying on principal component analysis of a minimum dataset ($MDS_{PCA}$) for assessing SQ, typically apply a single scoring and indexing method. It has been demonstrated that using multiple linear and nonlinear scoring functions with $MDS_{PCA}$ resulted in SQI values ranging from 0.34 to 0.93 (higher values assumed to have better soil quality), with up to 59% variability under similar soil conditions [17]. The use of additive versus weighted additive indexing methods with $MDS_{PCA}$ improved results to a range of 2–7% variability in SQI values. These findings highlight the inherent inconsistencies associated with existing scoring functions, indexing approaches, and data-set selection, as different methodological combinations often yield divergent SQI outcomes.

MDS frameworks based on inductive soil properties have been developed using statistical tools such as regression analysis and PCA [12]. However, these approaches tend to be highly site-specific. Historically, crops yield a deductive outcome of soil performance has been suggested as a qualitative measure of SQ [19]. Despite this, crop yield has rarely been incorporated in MDS selection for quantitative SQ assessment [20]. SQI calculated SQI using linear scoring and three indexing methods was strongly correlated with crop yield (r = 0.65 to 0.79) [21]. Nevertheless, no prior study has directly developed an MDS based on crop yield data, and concerns persist that crop yield can be maximized at the expense of sustaining soil quality. The inherent complexity and variability of soil systems, as well as the confounding influence of fertilization, irrigation, and other agronomic practices on crop yield often leads to misinterpretation of soil quality status when yields are used as the sole or primary indicator [22].

The SQ assessment process typically follows three key steps: (i) selection of soil indicators, either from the TDS or an MDS; (ii) scoring of indicators using linear or nonlinear functions; and (iii) integration of scores into a composite SQI [6,16,23,24]. While a range of scoring functions exists [10,25–30], two general scoring approaches are commonly employed: linear and nonlinear transformations. These are then integrated using one of three indexing methods: additive (SQIa), weighted additive (SQIw), or Nemoro index (SQIn) [1,6,15,24,31].

Linear scoring methods are straightforward and require only basic knowledge of threshold values, making them practical for many applications. In contrast, nonlinear scoring functions when informed by robust threshold data may provide more meaningful assessments by better capturing nonlinear soil function responses [15]. In contrast, the present study found that linear scoring with threshold limits combined with additive indexing produced the most consistent and dependable SQI values across the evaluated U.S. sites. This apparent discrepancy can be attributed to differences in soil types, climatic conditions, cropping systems, indicator selection strategies, and the inclusion of crop yield as a deductive criterion for minimum dataset development [18]. However, few studies have applied and compared both linear and nonlinear scoring methods alongside multiple indexing techniques under consistent soil conditions. SQI was assessed using $MDS_{PCA}$ with three linear and two nonlinear scoring methods, along with two indexing approaches [17]. In a similar study, both TDS and $MDS_{PCA}$ were applied in combination with one linear and one nonlinear scoring method and three indexing techniques [10]. These studies reported up to 50% variation in SQI values depending on the dataset, scoring function, and indexing method even under comparable conditions. Notably, nonlinear scoring methods produced higher SQI values in one case [10] whereas the opposite trend was observed in another [17]. These findings suggest that the performance of scoring and indexing approaches is context dependent, and that no single scoring–indexing combination is universally optimal. While nonlinear weighted additive indexing may enhance sensitivity in certain agroecosystems, particularly under short-term rotational systems, linear additive frameworks may offer greater robustness and transferability when applied across diverse soils and regions, as demonstrated in this study. In both cases, soil indicators were normalized using literature-based thresholds, which are often unavailable or site-specific. This underscores the need to develop location-specific threshold limits for accurate and dependable SQ assessment. But location-specific thresholds also limit the widespread application of such scoring methods.

In this study, we propose a comprehensive evaluation of four linear and two nonlinear scoring functions both with and without normalization using threshold values combined with three indexing methods to assess SQ. By explicitly comparing additive, weighted additive, and Nemoro indexing approaches under both linear and nonlinear scoring frameworks, this study aims to reconcile recent evidence favoring nonlinear weighted additive indexing with broader cross-site evaluation across contrasting soils and management systems. These methodologies will be calibrated and validated across several geographic locations. Additionally, we introduce a novel approach for MDS selection based on crop yield data, incorporating threshold limits for soil indicators to enhance the robustness and transferability of SQ assessment.

The specific objectives are to: (i) initial assessment of SQ of an experimental site in Indiana, USA, as a reference, using six scoring functions (four linear and two nonlinear), three indicator selection methods (TDS, $MDS_{PCA}$, and $MDS_{Corr}$), and three SQ indexing methods (SQIa, SQIw, and SQIn); (ii) identify and optimize the most effective combination of dataset,

scoring, and indexing methods for SQI evaluation; and (iii) calibrate and validate the optimized methods by assessing the SQ of experimental sites in Hoytville and Piketon, Ohio, and Shorter, Alabama, USA.

## Materials and methods

### Data source

Data on soil biological, chemical, and physical properties, and corn yields from replicated field experiments conducted at four experimental sites representing a range of soil types and climatic conditions in the United States [32,33]. The sites include Shorter, Alabama (E.V. Smith Research Center); Farmland, Indiana (Davis Purdue Agricultural Center); Hoytville, Ohio (Northwest Agricultural Research Station); and Piketon, Ohio (The Ohio State University South Centers research farm) [32].

A total of 576 geo-referenced composite soils were analyzed, with 144 samples collected from each site at 0–15 and 15–30 cm depth increments and were summed to represent the 0–30 cm soil profile for SQI calculations. Details of the soil sampling, processing and analysis of soil y properties are published or reported elsewhere [32,33].

### Soil quality index computations

To assess SQ using multi-scoring and indexing methods, 22 soil properties as potential SQ indicators were selected. A total of fifty-four (n = 54) SQI values were calculated for each location using multi-scoring (data normalization) and indexing (integration) approaches (Fig 1). For data normalization, four linear scoring methods (LSMs) and two non-linear scoring methods (NLSMs) were employed. The SQ indicator values were transformed to a common scale ranging between > 0 and < 1.0 using these LSM and NLSM methods [12].

The three types of datasets utilized were the TDS, $MDS_{Corr}$, and $MDS_{PCA}$. Subsequently, three SQ indexing methods were applied to the datasets that included additive-based (SQIa), weighted additive-based (SQIw), and Nemoro-based (SQIn). Details of the data normalization (scoring) and indexing equations are provided in Table 1 and Table 2.

The assessment of the SQI followed a logical sequence: (i) selecting soil properties and quotients as potential indicators, (ii) transforming these indicators into normalized scores, and (iii) integrating the scores into a composite index [15]. Each step involved in the SQI assessment process is illustrated in Fig 1.

### Indicator selection for soil quality assessment

**Total dataset (TDS).** Twenty-one soil properties and quotients were included in the TDS as inductive indicators comprising 11 chemical, 8 physical, and 2 biological indicators. The chemical indicators included non-microbial biomass carbon (NSMB), pH, electrical conductivity (ECe), total nitrogen (TN), total organic carbon (SOC), active carbon (AC), nitrogen pool index (NPI), carbon pool index (CPI), carbon lability (CL), carbon lability index (CLi), and carbon management index (CMI). The 8 physical indicators consisted of bulk density (ρb), macroaggregate stability (MaAS), microaggregate stability (MiAS), aggregate stability (AS; calculated as the sum of MaAS and MiAS), mean weight diameter (MWD), geometric mean diameter (GMD), aggregation index (AI), and aggregate persistence index (PI). The biological indicators included total microbial biomass (SMB) and the metabolic quotient (SMB:SOC as qR). The soil properties and indicators have been recommended as measured or divided SQ indicators due to their strong relationships with soil functions and agroecosystem services [5,10,12,34,35,36].

**Selection of minimum data set based on correlation (MDS_Corr).** Based on Pearson's correlation coefficients between corn yield (deductive indicator) and inductive soil indicators in the control treatment (corn without any gypsum amendments and cover crops) across all experimental locations, $MDS_{Corr}$ indicators were selected (Table 3). SQ indicators with correlation coefficients greater than 0.5 and significant at p < 0.01 were chosen from the TDS. The indicators TN (0.61), SOC (0.55), AC (0.57), MaAS (0.56), MWD (0.55), and GMD (0.6) showed positive and significant correlations

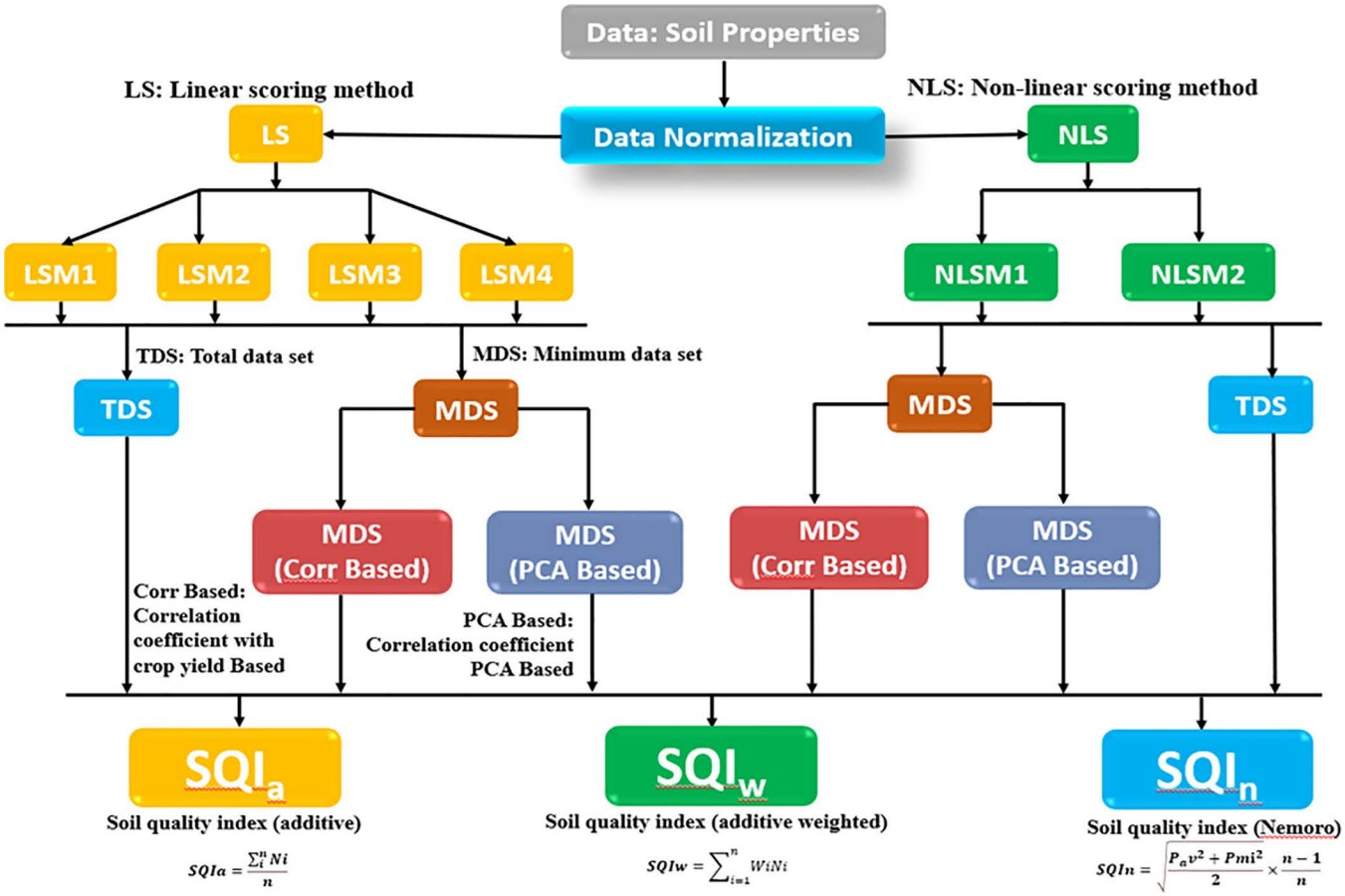

**Fig 1. A conceptual model for computing soil quality indices.** [DS: Dataset; TDS: Total dataset; MDS: Minimum dataset; LS: Linear scoring; NLS: Non-linear scoring; LSM: Linear scoring method; SQIa: Soil quality index (additive method); SQIw: Soil quality index (weighted additive method); SQIn: Soil quality index (Nemoro method)].

with corn yield, while $\rho b$ (−0.61) and MiAS (−0.62) were negatively correlated. In total, eight SQ indicators were retained in the MDS$_{Corr}$. To evaluate potential multicollinearity among the selected indicators, variance inflation factors (VIF) were evaluated for the MDS$_{Corr}$ variables. Although few indicators describing soil aggregation (e.g., MWD and GMD) were moderately correlated, VIF values remained below commonly accepted thresholds (VIF < 5), indicating that multicollinearity was not significant to warrant variable exclusion. Moreover, MWD and GMD were retained because they describe complementary aspects of soil aggregate size distribution rather than redundant information and have been shown to respond differently to management and soil texture in previous studies [4,6].

The corn yield and associated SQ parameters from the Indiana site, Hoytville (Ohio) site, Alabama site, and Piketon (Ohio) site are presented in S1–S5 Tables). Among the positively correlated indicators, TN and GMD exhibited stronger correlations with corn yield than SOC, AC, MaAS, and MWD, suggesting that TN and GMD are the most influential SQ indicators for corn production. However, it is important to note that a higher TN value does not necessarily imply improved soil or environmental quality, especially if achieved through excessive nitrogen fertilization. While increasing TN may enhance short-term yield, it may contribute to runoff, leaching, greenhouse gas emissions, and soil acidification. Conversely, the negative correlations of $\rho b$ and MiAS with corn yield suggest that soil structural conditions associated with higher soil compaction with reduced porosity may constrain corn productivity. However, $\rho b$ alone does not uniquely

**Table 1. Scoring equations and method used for normalization of soil indicators data.**

| Scoring Eq. | Equation | Scoring method | References |
|---|---|---|---|
| **Linear** | $Y = \left(\frac{X}{X\,Max}\right)$ More is better.<br>$Y = \left(\frac{X}{X\,Min}\right)$ Less is better.<br>*X: Soil property value; X Max: Maximum value or upper threshold value; X Min: Minimum value or lower threshold value* | Scoring method 1 (LSM1) | [24] |
| | $Y = \left(\frac{X}{X\,Max}\right)$ More is better.<br>$Y = 1 - \left(\frac{X}{X\,Min}\right)$ Less is better.<br>*X: Soil property value; X Max: Maximum value or upper threshold value; X Min: Minimum value or lower threshold value* | Scoring method 2 (LSM2) | [33] |
| | $Y = 0.1 + \left(\frac{X-b}{a-b}\right) * 0.9$ More is better.<br>$Z = 1 - \left(\frac{X-b}{a-b}\right) * 0.9$ Less is better.<br>*X: value of the variable to transform; a and b: maximum and minimum value or upper and lower threshold values of the variable, respectively* | Scoring method 3 (LSM3) | [26] |
| | $Y = \left(\frac{X-S}{t-S}\right)$ More is better.<br>$Y = 1 - \left(\frac{X-S}{t-S}\right)$ Less is better.<br>*X: value of the variable to transform; S and t: maximum and minimum value or upper and lower threshold values of the variable, respectively* | Scoring method 4 (LSM4) | [16,17,27] |
| **Non-linear** | $f(x) = \frac{1}{\delta\sqrt{2\pi}} e^{-\frac{(x-\mu)^2}{2\sigma^2}}$<br>$\infty < x < \infty$<br>*Cumulative normal distribution function (CND)*<br>*was used to score variables by using estimated means (m) and standard deviations (s). To get the cumulative normal probabilities and score all measured values, the NORMDIST function in Excel (Microsoft Office 2010) was used.* | Scoring method 1 (NLSM1) | [15,28] |
| | $f(x) = \frac{a}{1+\left(\frac{x}{x_0}\right)^b}$<br>*a: maximum score equal to 1, x: soil variable value, $X_0$: mean value of the variable and b: slope; it assumed to be −2.5 for 'more is better' functions and + 2.5 for 'low is better' ones.* | Scoring method 1 (NLSM2) | [10] |

**Table 2. Integration equations and method used for the calculation of the soil quality index.**

| Integration | Integration Equation | Index | Reference |
|---|---|---|---|
| Additive | $SQIa = \frac{\sum_i^n Ni}{n}$<br>*Ni are the indicator scores, and n is the number of indicators.* | $SQI_a$ | [15,30] |
| Weighted additive | $SQIw = \sum_{i=1}^{n} WiNi$<br>*Wi is the weighting factor for the soil indicator derived from the factor analysis and Ni is the indicator score.* | $SQI_w$ | [1] |
| Nemoro | $SQIn = \sqrt{\frac{P_a v^2 + Pm^2}{2}} \times \frac{n-1}{n}$<br>*Pav is the average and Pmi is the minimum scores of the selected indicators at each sampling point.* | $SQI_n$ | [10] |

diagnose soil compaction, particularly across contrasting soil textures, as soils with similar pb values may differ in pore connectivity, aeration, and root growth resistance. Accordingly, ρb is interpreted as a descriptive physical indicator of soil structure rather than a direct functional measure of compaction. Therefore, the selection of three chemical SQ (TN, SOC, and AC) and five physical (pb, MaAS, MiAS, MWD, and GMD) indicators were retained for use in the MDS$_{Corr}$, with the recognition that ρb serves as a supporting structural descriptor whose interpretation depends on soil texture and site-specific conditions.

**Selection of minimum dataset based on principal component analysis (MDS$_{PCA}$).** The MDS$_{PCA}$ was selected by applying PCA (OriginPro®) to the TDS to reduce data redundancy and identify the core indicators without relying on

**Table 3. Selection of minimum dataset soil indicators (MDSCorr) based on Pearson correlation between soil properties and corn productivity.**

| Soil properties | Correlation coefficient | P value | MDS$_{Corr}$ |
|---|---|---|---|
| SMB (mg/kg) | 0.03 | 0.82 | |
| Non-SMB (%) | **0.55** | 0.00 | |
| qR (%) | −0.26 | 0.04 | |
| pH | 0.44 | 0.00 | |
| ECe (µS/cm) | 0.17 | 0.18 | |
| TN (%) | **0.61** | 0.00 | TN |
| SOC (%) | **0.55** | 0.00 | SOC |
| AC (mg/kg) | **0.57** | 0.00 | AC |
| NPI | −0.07 | 0.59 | |
| CPI | −0.01 | 0.95 | |
| CL | −0.06 | 0.63 | |
| Cli | 0.18 | 0.14 | |
| CMI | 0.42 | 0.00 | |
| nCMI | 0.20 | 0.12 | |
| ρb (g/cm³) | **−0.61** | 0.00 | ρb |
| MaAS (%) | **0.56** | 0.00 | MaAS |
| MiAS (%) | **−0.62** | 0.00 | MiAS |
| AS (%) | 0.45 | 0.00 | |
| SI | 0.35 | 0.00 | |
| PI (%) | 0.44 | 0.00 | |
| MWD (mm) | **0.55** | 0.00 | MWD |
| GMD (mm) | **0.60** | 0.00 | GMD |

Indicators included in MDSCorr were selected based on |r| ≥ 0.5 and p < 0.01.

SMB: soil microbial biomass; Non-SMB: non-soil microbial biomass; qR: SMB: SOC; ECe: electric conductivity of soil; TN: total nitrogen; SOC: Soil organic carbon; AC: active carbon; NPI: nitrogen pool index; CPI: carbon pool index; CL: carbon lability; Cli: carbon lability index; CMI: carbon management index; nCMI: normalized carbon management index; ρb: soil bulk density; MaAS: macroaggregate stability; MiAS: microaggregate stability; AS: total aggregate stability; SI: stability index; and PI: persistent index, MWD: mean weight diameter; GMD: geometric mean diameter.

subjective judgments or literature-based values. By reducing dimensionality, multiple indicators are consolidated into a smaller set of independent indicators, thereby eliminating autocorrelations among the original variables.

Principal components (PCs) with eigenvalues ≥1 were extracted, and variables with absolute factor loadings ≥0.5 within the same PC were grouped together. Highly weighted indicators were defined as those with absolute factor loading values ≥0.5. If an indicator's loading value was less than 0.5 for all PCs, it was assigned to the group where it exhibited the highest loading.

Following grouping, the norm value was calculated for each evaluation indicator. It has been noted that eigenvectors within a PC do not provide information about the magnitude of a variable in the multi-dimensional space, either for PCs or for the original variables [37]. Therefore, relying solely on loading might exclude important indicators. The norm represents the magnitude (length) of the vector in the multi-dimensional space spanned by the selected PCs and reflects the composite contribution of each indicator across all PCs. The norm value was calculated using the following formula:

$$N_{ik} = \sqrt{\sum_{i=1}^{k} (u_{ik}^2 \lambda_k)}$$

(1)

Where, $N_{ik}$ is the combined loading of the *ith* indicator on the first k principal components with eigenvalues ≥1, *Uik* is the loading of the *ith* indicator on the k*th* principal component, λk is the eigenvalue of the k*th* principal component.

For each group, indicators within 10% of the highest norm values were selected. If significant correlations existed between indicators within a group, the indicator with the highest norm value was included in the MDS. If non-significant correlations were observed, all indicators within the group were included [38]. The PCA and correlation data are presented in S6-S9 Tables.

For the Indiana site, the PCs were PC1 (10.1)> PC2 (3.8)> PC3 (3)> PC4 (1.9), which explained 78.8% of the total variance. Based on PCA and correlation analyses (S6 and S6(a) Tables), SMB, Non-SMB, AS, and PI were selected for the MDS_PCA. For Hoytville (Ohio north) site, the PCs were PC1 (9.4)> PC2 (5.3)> PC3 (2.7)> PC4 (2.2)> PC5 (1.3), accounting for 86.9% of the total variance. Based on results (S7 and S7(a) Tables), qR, pH, AC, NPI, and MiAs were included in the MDS_PCA. For Alabama site, the PCs were PC1 (7.5)> PC2 (5.1)> PC3 (3.6)> PC4 (2.5)> PC5 (1.2), accounting for 83.1% of the total variance. After evaluating correlation coefficients and p-values (S8 and S8(a) Tables), the indicators SOC, NPI, Cli, CMI, pb, AS, and GMD were selected for the MDS_PCA. For Piketon (Ohio south) site, the PCs were PC1 (12.7)> PC2 (4.5)> PC3 (2.7)> PC4 (1.2), with a cumulative variance contribution of 88.3%. Based on PCA, correlation, and p-value analyses (S9 and S9(a) Tables), pH, NPI, CMI, nCMI, and PI were selected for the MDS_PCA used to assess soil quality. These findings are consistent with previous methodologies [1,15,35].

**Evaluation of soil quality indexing methods.** The SQ indexing methods were evaluated by sensitivity analysis [16]. The equation for the calculation is as follows:

$$Sensitivity\ (S) = \frac{SQI\ (max)}{SQI\ (min)}$$

(2)

Where SQI (max) and SQI (min) are the maximum and minimum SQ values obtained under each combination of scoring function (e.g., linear, non-linear) and indexing method (e.g., additive, weighted additive, PCA-based) applied to each dataset selection method (e.g., TDS, MDS_PCA, MDS_Corr). The SQI values were computed using standardized scores for the selected soil indicators, followed by index computation according to the respective method. It is important to note that this sensitivity metric does not measure reproducibility, precision, or variation of repeated results. Instead, it quantifies the responsiveness of each soil quality indexing method by comparing the spread between maximum and minimum SQI values under different scoring and dataset selection approaches. A higher sensitivity value indicates greater discriminative ability of the indexing method to capture management-induced differences in soil quality.

**Accuracy verification of soil quality indexing with minimal dataset.** The Nash effective coefficient ($E_f$) and relative deviation coefficient ($E_R$) were used to evaluate the accuracy of the MDS [39,40]. The formulas for the calculations are as below:

$$E_f = 1 - \frac{\sum (R_0 - Rcal)^2}{\sum (R_0 - \overline{R}_0)^2}$$

(3)

$$E_R = \left| \sum_{i=1}^{n} R_{0\ i} - R_{cali} \right| / \sum_{i=1}^{n} R_{0i}$$

(4)

Where $R_0$ and $\overline{R}_0$ are the SQI value and mean SQI, respectively. Those SQI were calculated based on TDS. $R_{cal}$ was the value of SQI calculated with MDS, and $i$ is the serial number.

**Development of threshold limit values for soil quality indicators.** To develop threshold limit values for each experimental site, the cumulative normal distribution (CND) method was employed [28]. Soil data from the control treatment, which was defined as corn under no-till management without any gypsum applications and cover crops, were

first transformed into a cumulative normal distribution using the NORMDIST function in Excel (Microsoft® Office 2010). The resulting CND data were then converted into a > 0 to < 100 scoring system. These scores were used to construct non-linear curves, which were fitted using non-linear curve fitting equations with OriginPro®. The shapes of the resulting scoring curves were characterized into three types, following [41]:

1. Bell-shaped, where the maximum score represents the optimal range of a given indicator and both lower and higher values may indicate suboptimal conditions.

2. Sigmoid with an upper asymptote, where higher indicator values are associated with improved SQ (i.e., more is better); and

3. Sigmoid with a lower asymptote, where lower indicator values are preferred (i.e., less is better).

From the plotted curves (Fig 2), specific thresholds were identified for each SQ indicator. The lower threshold (LT) corresponds to a score of 0, representing the poorest SQ or extremely limiting condition. The critical threshold (CT) also referred to as the baseline threshold (BT) is set at a score of 50, indicating a moderate or transitional SQ level. The upper threshold (UT) corresponds to a score of 100, representing the most optimal soil condition based on the fitted curve. The final threshold values for each indicator at the Indiana site (as a reference) are presented in Table 4. Threshold values for the other geographic sampling sites of Hoytville (north Ohio), Alabama, and Piketon (southern Ohio) are provided in S10-S12 Tables respectively.

## Results and discussions

### Variability of soil quality indices with different datasets, and scoring and indexing methods

The results show that the SQI varied widely depending on the combination of datasets, including TDS, MDS$_{Corr}$, and MDS$_{PCA}$, as well as the scoring methods (LSM1, LSM2, LSM3, LSM4, NLSM1, NLSM2) and indexing methods (SQIa, SQIw, and SQIn) used (Fig 3). The calculation of the soil quality index (SQIw) for the Indiana reference site using TDS and MDS$_{Corr}$ without threshold values is presented in S13 and S14 Tables, respectively. For the Indiana site (as a reference), the SQ indicators were normalized both with and without the use of threshold limit values. The SQI ranged from 0.33 to 0.6 when no threshold limit was applied and from 0.32 to 0.67 when threshold limits were used. The difference between the lowest and highest SQI scores for Indiana site ranged from 27% to 35%, indicating that dataset selection,

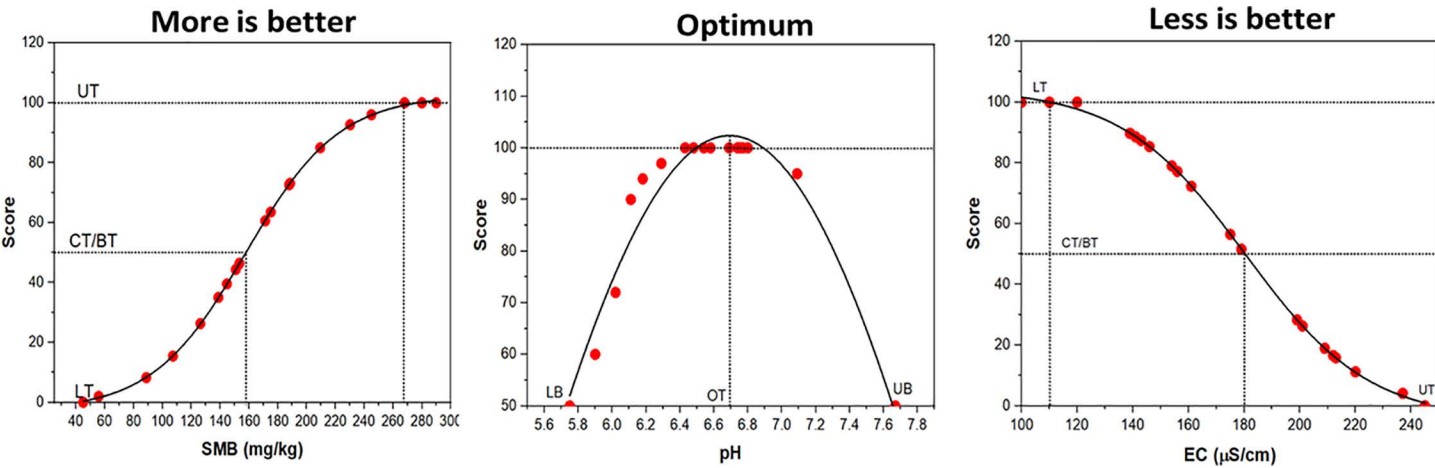

**Fig 2. Determination of threshold values of soil quality indicators.** (LT: Lower threshold; CT/BT: Critical/Baseline threshold; UT: Upper threshold).

**Table 4. Soil quality threshold values for Indiana site (as reference).**

| Soil properties | Lower threshold (LT) (score: 0%) | Critical or base threshold (CT or BT) (score: 50%) | Upper threshold (UT) (score: 100%) | Optimum threshold (OT) (score:100%) | Scoring curve |
|---|---|---|---|---|---|
| SMB-C (mg/kg) | 45 | 158 | 268 | | More is better |
| Non-SMB-C (%) | 0.6 | 1.35 | 2.1 | | More is better |
| qR (%) | 0.4 | 1.15 | 2.28 | | More is better |
| pH | 5.75 | | 7.64 | 6.7 | Optimum |
| ECe (μS/cm) | 113 | 180 | 245 | | Less is better |
| TN (%) | 0.08 | 0.15 | 0.22 | | More is better |
| SOC (%) | 0.5 | 1.4 | 2.1 | | More is better |
| AC (mg/kg) | 220 | 560 | 840 | | More is better |
| NPI | 0.7 | 1.34 | 2.25 | | More is better |
| CPI | 0.5 | 1.35 | 2.0 | | More is better |
| CL | 0.02 | 0.04 | 0.06 | | More is better |
| Cli | 0.7 | 1.18 | 1.8 | | More is better |
| CMI | 0.5 | 1.57 | 2.3 | | More is better |
| nCMI | 30 | 73 | 110 | | More is better |
| ρb (g/cm³) | 1.05 | 1.30 | 1.55 | | Less is better |
| MaAS (%) | 50 | 58 | 72 | | More is better |
| MiAS (%) | 1.0 | 6.8 | 11.5 | | Less is better |
| AS (%) | 54.5 | 65.8 | 73 | | More is better |
| SI | 1.0 | 12 | 55 | | More is better |
| PI (%) | 5.5 | 14 | 25 | | More is better |
| MWD (mm) | 0.4 | 1.27 | 2.3 | | More is better |
| GMD (mm) | 0.53 | 1.03 | 1.83 | | More is better |

SMB: soil microbial biomass; Non-SBM: non-microbial biomass carbon; qR: SMB: SOC; ECe: electric conductivity; TN: total nitrogen; SOC: Soil organic carbon; AC: active carbon; NPI: nitrogen pool index; CPI: carbon pool index; CL: carbon lability; Cli: carbon lability index; CMI: carbon management index; nCMI: normalized carbon management index; ρb: bulk density; MaAS: macroaggregate stability; MiAS: microaggregate stability; AS: total aggregate stability; SI: stability index; and PI: persistent index, MWD: mean weight diameter; GMD: geometric mean diameter.

scoring functions, and indexing approaches each exerted a strong influence on SQI outcomes. Overall, substantial variability among SQI values highlights the sensitivity of soil quality assessment to methodological choices.

As shown in Fig 3a, without threshold normalization, linear scoring methods (LSM1-LSM4) applied to the TDS and MDS_{Corr} datasets and integrated using additive (SQIa) or weighted additive (SQIw) indexing generally producing higher SQI values, with median SQI values commonly ranging between approximately 0.53 to 0.6. In contrast, nonlinear scoring methods (NLSM1 and NLSM2), regardless of dataset or indexing approach, yielded comparatively lower and more compressed SQI distributions, typically centered between 0.38 to 0.49. When threshold limits were applied (Fig 3b, overall SQI values increased modestly for both linear and nonlinear scoring methods, and the spread of SQI values was reduced, indicating improved consistency across dataset-indexing combinations. Linear scoring methods combined with SQIa or SQIw remained among the highest-performing approaches, whereas nonlinear scoring methods continued to exhibit narrower SQI ranges and lower median values. Collectively, the results in Fig 3a and Fig 3b demonstrate that SQI outcomes are sensitive to the choice of scoring function, dataset, and indexing method, and that no single approach performs optimally under all conditions. These findings reinforce earlier observations that SQ assessments are method-dependent and may vary across soils, datasets, and analytical frameworks, as reported in previous studies [6,26,42,43]. To reduce the disparity between the lowest and highest SQI scores at each sampling location, an optimized method for SQI calculation

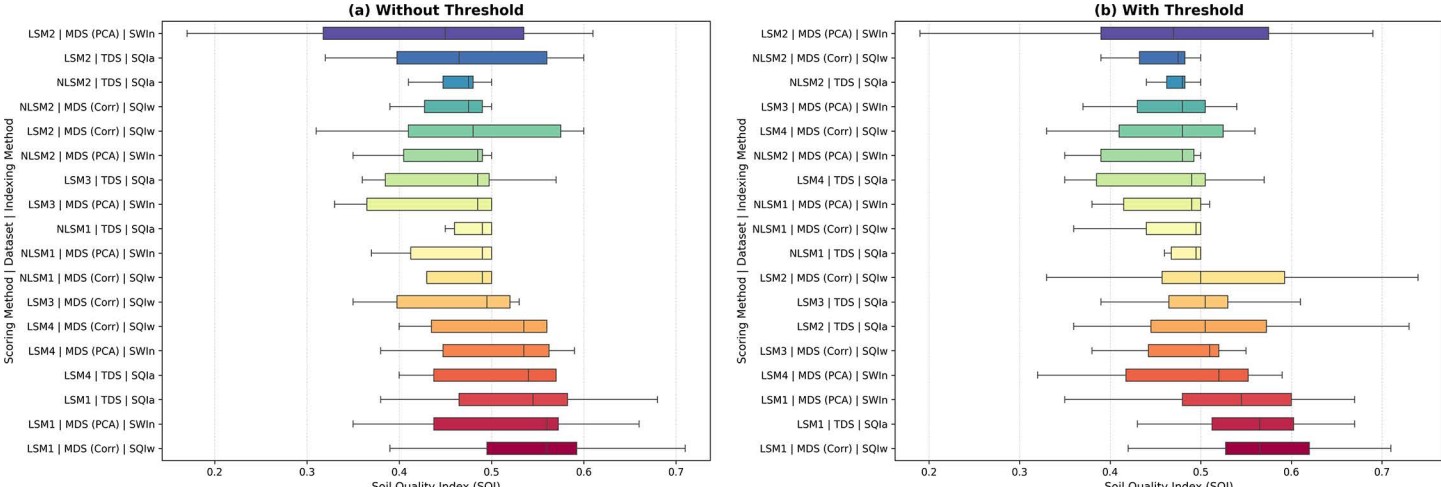

**Fig 3. Indiana Soil Quality Index (SQI) distributes across different method combinations under two scoring conditions: (a) without threshold and (b) with threshold. Each method combination consists of a scoring function, dataset selection approach, and indexing technique.** [LSM1: Linear scoring method 1; LSM2: Linear scoring method 2; LSM3: Linear scoring method 3; LSM4: Linear scoring method 4; NLSM1: Nonlinear scoring method 1; NLSM2: Nonlinear scoring method 2; TDS: Total dataset; MDSCorr: Minimum dataset selection based on correlation with crop yield; MDSPCA: Minimum dataset based on principal component analysis; SQIa: Soil quality index (additive); SQIw: Soil quality index (additive weighted); SQIn: Soil quality index (Nemoro), SQI: Soil quality index].

needs to be developed by systematically examining various datasets, scoring, and indexing methods. Recent studies conducted under different cropping systems have shown that nonlinear scoring functions combined with weighted additive indexing can improve the sensitivity and effectiveness of SQIs. For example, Iheshiulo et al. (2024) reported that a nonlinear weighted additive indexing approach outperformed alternative methods in capturing soil health responses under short-term crop rotations in the Canadian prairies [18]. In contrast, the present study found that linear scoring combined with additive or weighted additive indexing produced more consistent SQI values across the evaluated U.S. sites. This apparent difference reflects variations in soil types, climatic conditions, cropping systems, indicator selection strategies, and the incorporation of crop yield as a deductive criterion for minimum dataset development. Together, these findings indicate that the performance of scoring and indexing approaches is context dependent and that no single scoring–indexing combination is universally optimal.

## Optimization of soil quality indexing methods

To optimize the scoring methods, SQIs were evaluated using different datasets, scoring approaches, and indexing methods (Table 5). The results demonstrated notable variation in SQI values depending on the scoring and indexing methods applied, using the Indiana site as a reference. SQI values calculated with the SQIn method were significantly lower for both linear and nonlinear scoring across all three datasets compared to those derived from the SQIa and SQIw methods. Specifically, SQI values differed by 14% to 15% with SQIn when using linear scoring relative to SQIa and SQIw, and by 7% to 13% when comparing across the same indexing and scoring methods. In contrast, SQI values obtained with SQIa and SQIw differed by only 0% to 1% between linear and nonlinear scoring. Due to the substantial disparity in SQI values produced by the SQIn method for Indiana soils, SQIn, Nemoro indexing was omitted from further calculations to minimize redundancy (Fig 3). Similar behavior of the Nemoro index, which emphasizes minimum indicator values and often produces lower SQI scores, has been reported in earlier SQ assessments and may limit its applicability when multiple indicators exhibit contrasting responses [26,44].

**Table 5. Combination effects of dataset selection, scoring function, and indexing methods on soil quality assessment of Indiana site, USA (as reference).**

| Type of Dataset | Scoring Method | Indexing Method | No. of Scoring method | Mean SQI | SD | SEM | CV |
|---|---|---|---|---|---|---|---|
| TDS | LN | $SQI_a$ | 4 | 0.54 | 0.04 | 0.02 | 0.002 |
| | | $SQI_w$ | 4 | 0.55 | 0.04 | 0.02 | 0.002 |
| | | $SQI_n$ | 4 | 0.40 | 0.03 | 0.02 | 0.001 |
| | NLN | $SQI_a$ | 2 | 0.49 | 0.01 | 0.01 | 0.000 |
| | | $SQI_w$ | 2 | 0.49 | 0.01 | 0.01 | 0.000 |
| | | $SQI_n$ | 2 | 0.45 | 0.02 | 0.01 | 0.000 |
| $MDS_{Corr}$ | LN | $SQI_a$ | 4 | 0.56 | 0.04 | 0.02 | 0.002 |
| | | $SQI_w$ | 4 | 0.55 | 0.04 | 0.02 | 0.001 |
| | | $SQI_n$ | 4 | 0.42 | 0.03 | 0.02 | 0.001 |
| | NLN | $SQI_a$ | 2 | 0.49 | 0.01 | 0.01 | 0.000 |
| | | $SQI_w$ | 2 | 0.49 | 0.01 | 0.01 | 0.000 |
| | | $SQI_n$ | 2 | 0.42 | 0.01 | 0.01 | 0.000 |
| $MDS_{PCA}$ | LN | $SQI_a$ | 4 | 0.53 | 0.03 | 0.01 | 0.001 |
| | | $SQI_w$ | 4 | 0.53 | 0.03 | 0.01 | 0.001 |
| | | $SQI_n$ | 4 | 0.36 | 0.07 | 0.03 | 0.005 |
| | NLN | $SQI_a$ | 2 | 0.49 | 0.00 | 0.00 | 0.000 |
| | | $SQI_w$ | 2 | 0.49 | 0.00 | 0.00 | 0.000 |
| | | $SQI_n$ | 2 | 0.36 | 0.01 | 0.01 | 0.000 |

TDS: total dataset; $MDS_{Corr}$: minimum dataset selected based on correlation with corn yield; $MDS_{PCA}$: minimum dataset selected using principal component analysis; LN: linear scoring method; NLN: nonlinear scoring method; $SQI_a$: additive soil quality index; $SQI_w$: weighted additive soil quality index; $SQI_n$: Nemoro soil quality index.

Nonlinear scoring methods produced SQI values that were 4% to 6% lower compared to linear methods (Table 5). Additionally, the difference between SQI values calculated using SQIa and SQIw with nonlinear scoring was small (0% to 1%) compared to that observed with linear methods. While it is difficult to draw a definitive conclusion regarding the superiority of linear or nonlinear scoring methods for achieving consistent SQI scores with both SQIa and SQIw indexing, the results suggest that both SQIa and SQIw are effective for SQI calculation.

## Optimization of soil quality scoring and normalization methods

To examine the effects of four linear scoring methods and two nonlinear scoring methods on SQI assessment, several radar graphs were generated using data from the Indiana site as a reference (Fig 4). The normalized scores for each SQ parameter varied significantly across the linear scoring methods (LSM1, LSM2, LSM3, and LSM4); however, these variations were not uniform and depended on whether threshold limits were applied. The differences between the minimum and maximum scores among SQ indicators ranged from 10% to 50%. Additionally, a 7% difference was observed between threshold and no-threshold linear scoring methods. In contrast, the normalized scores obtained with the nonlinear scoring methods (NLSM1 and NLSM2) were mostly uniform, with minimal differences across all SQ parameters, ranging from 0.5% to 4%. Only negligible differences (0.2%) were observed between threshold and non-threshold nonlinear scoring methods. These results suggest that linear scoring methods contribute more to the variability of SQI compared to nonlinear methods (Fig 3). Previous studies have similarly shown that nonlinear scoring functions tend to compress indicator values within narrower ranges, whereas linear scoring preserves greater variability and sensitivity among SQ indicators [15,45,46].

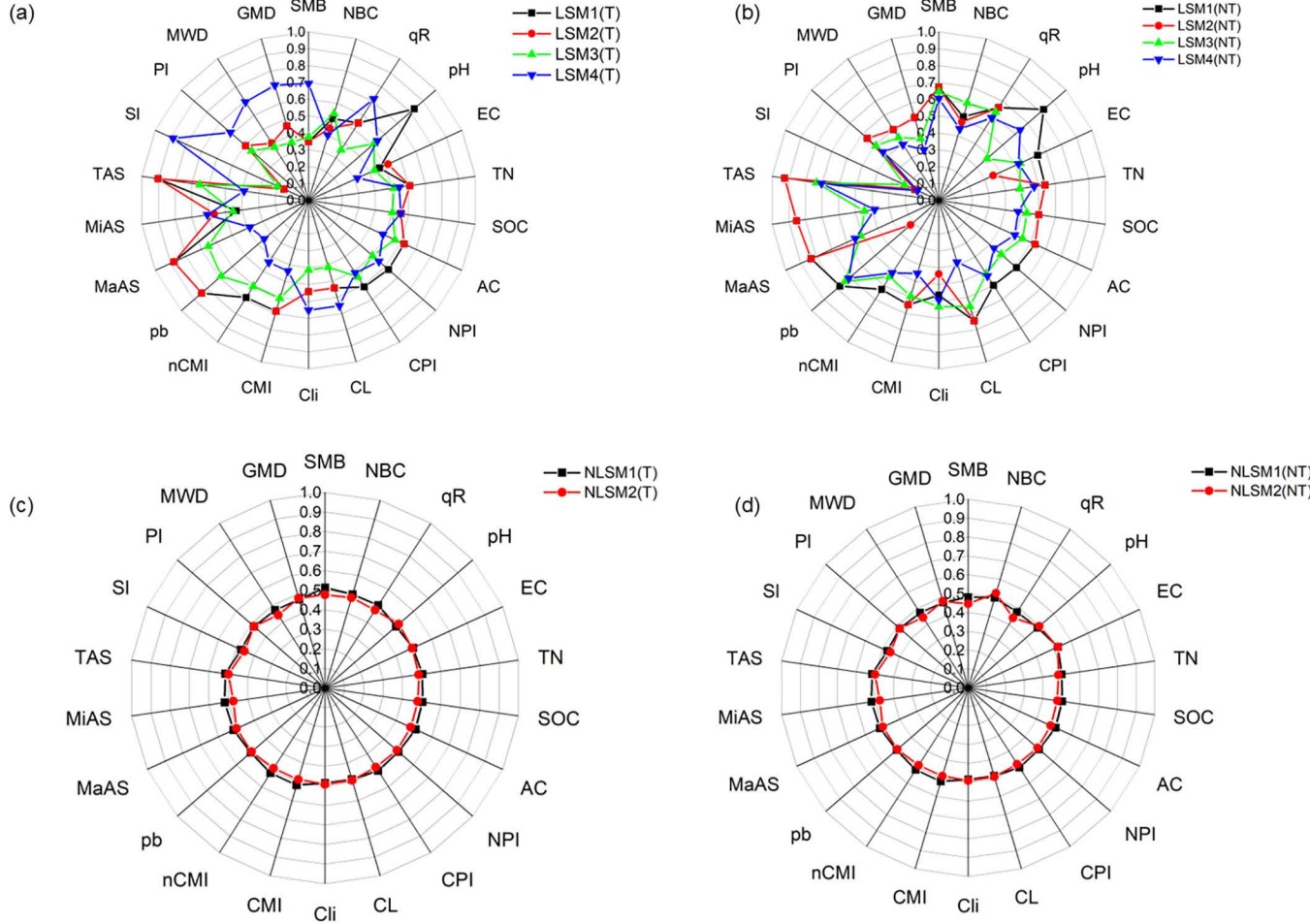

**Fig 4. Radar graphs created using the variation of normalized values (Indiana site as a reference) by applying four linear and two non-linear scoring methods.** [(T: Treshold value used and NT: Threshold value not used), SMB: total microbial biomass; Non-SMB: non-microbial biomass; qR: SMB: SOC; ECe: electric conductivity; TN: total nitrogen; SOC: Soil organic carbon; AC: active carbon; NPI: nitrogen pool index; CPI: carbon pool index; CL: carbon lability; Cli: carbon lability index; CMI: carbon management index; nCMI: normalized carbon management index; ρb: bulk density; MaAS: macroaggregate stability; MiAS: microaggregate stability; AS: total aggregate stability; SI: stability index; and PI: persistent index, MWD: mean weight diameter; GMD: geometric mean diameter].

To assess the similarity among the different scoring methods used for normalizing SQ parameters, Pearson's correlation coefficient (r) was evaluated among the four linear and two nonlinear scoring methods (Fig 5). Using LSM1 as the reference method, strong positive correlations were observed with LSM2 (r = 0.94) and LSM3 (r = 0.94), indicating close agreement among these linear scoring approaches. In contrast, LSM1 exhibited a strong negative correlation with LSM4 (r = −0.72), reflecting an inverse response between these two linear methods. The nonlinear scoring methods (NLSM1 and NLSM2) showed weak correlations with LSM1 (r = 0.52 and −0.5, respectively), indicating substantial methodological differences between linear and nonlinear normalization approaches. Consistent with these relationships, coefficients of determination from linear regression analysis show that LSM2 and LSM3 explain a considerable proportion of the variability in LSM1 ($R^2$ = 0.96 and 0.88, respectively), whereas LSM4 and the nonlinear methods explain less variance (Fig 5). Based on these results, three linear scoring methods (LSM1, LSM2, and LSM3) and one nonlinear scoring method (NLSM1) were selected for subsequent SQI calculations to balance methodological consistency and functional contrast.

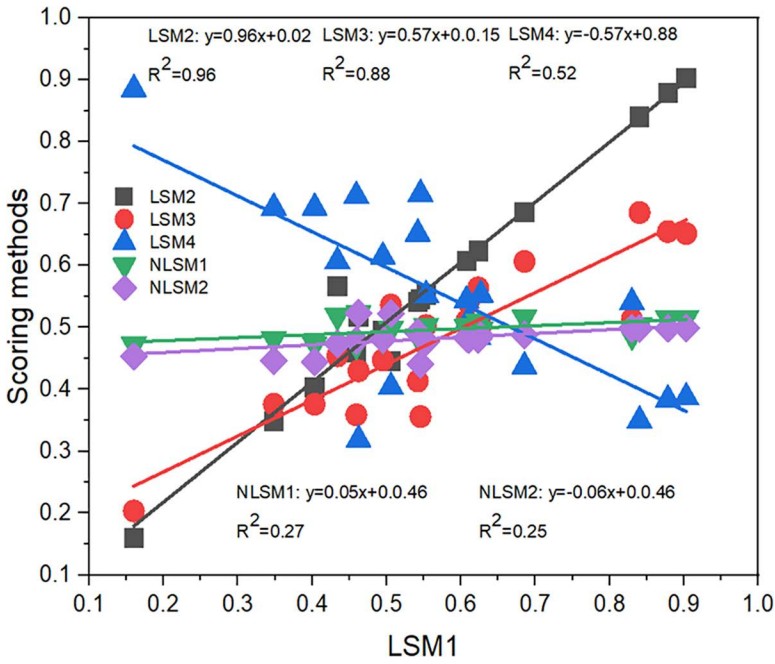

**Fig 5. Linear regression among all the scoring methods and their relationships.** (LSM1: Linear scoring method 1; LSM2: Linear scoring method 2; LSM3: Linear scoring method 3; LSM4: Linear scoring method 4; NLSM1: Nonlinear scoring method 1; and NLSM2: Nonlinear scoring method 2).

## Correlation based minimum dataset and its relationship with total dataset

The TDS includes all measured SQ indicators and provides a comprehensive basis for SQI calculation. While $MDS_{PCA}$ are commonly used, we developed $MDS_{Corr}$ derived from the relationship between soil indicators and crop yield (e.g., corn productivity from control treatment across all sites). To evaluate the performance of $MDS_{Corr}$, relationships between TDS and $MDS_{Corr}$, as well as between TDS and $MDS_{PCA}$, were examined using the Indiana site as a reference (Fig 6). Analyses incorporated averages of three linear and two nonlinear scoring methods combined with two indexing approaches.

Under linear scoring, $MDS_{Corr}$ exhibited a stronger and more consistent relationship with TDS ($R^2 = 0.53$) than $MDS_{PCA}$ ($R^2 = 0.27–0.70$), reflecting improved alignment across a broader range of SQI values. In contrast, nonlinear scoring methods compressed SQI values into a narrower range, producing clustered data distributions with limited mid-range representation (Fig 6). As a result, the high coefficients of determination observed under nonlinear scoring (up to $R^2 = 0.93$ for $MDS_{Corr}$) primarily reflect agreement among clustered endpoint values rather than predictive relationships across the full SQI domain. Overall, these results indicate that $MDS_{Corr}$ is more consistent with TDS and better suited than $MDS_{PCA}$ for SQI assessment, particularly under linear scoring conditions where SQI variability is better resolved and regression relationships are more informative. Similar limitations of PCA-based minimum datasets, including site-specific indicator selection and reduced transferability, have been widely reported in the SQ literature [6,20,47].

## Selection of minimum dataset, scoring, and indexing methods for soil quality

While the correlation results discussed above did not directly identify the most suitable scoring methods, they instead underscored the importance of appropriate dataset selection. Consequently, sensitivity analyses were conducted using the Indiana site as a reference (Table 6). In these analyses, the SQI method with the highest sensitivity value was considered preferable, as greater sensitivity indicates a stronger response to perturbations and management practices [41].

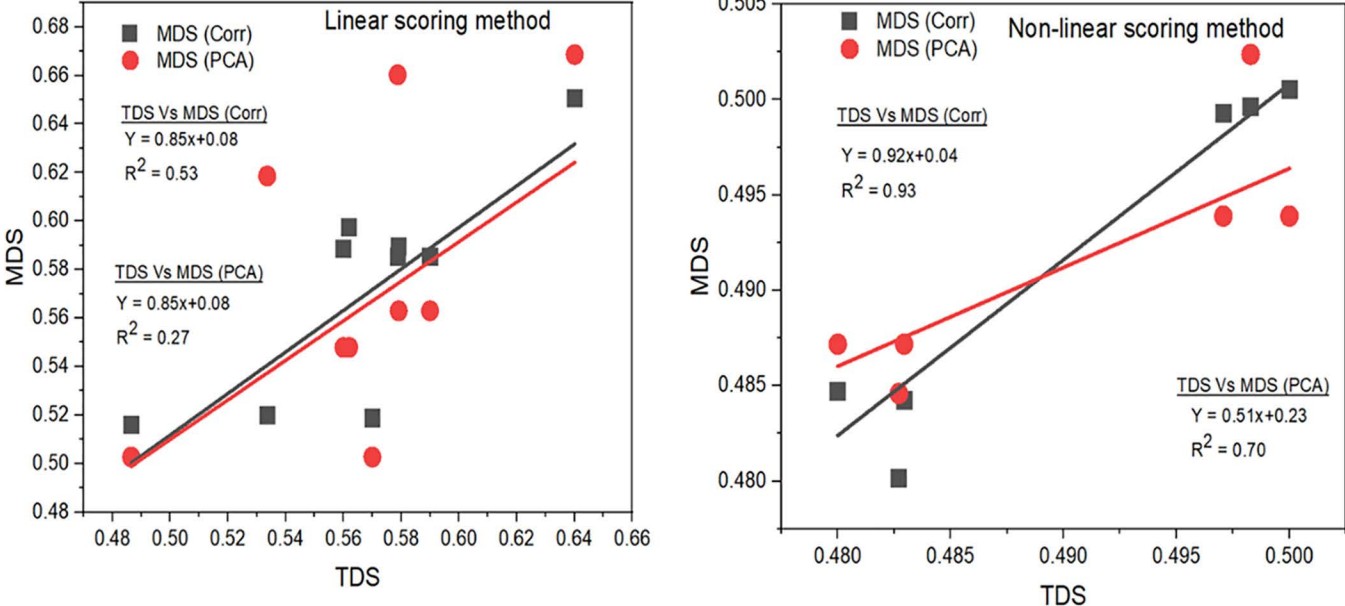

**Fig 6. Correlation of minimum dataset soil quality indicators.** [(MDS_Corr and MDS_PCA with total datasets (TDS) of soil quality indicators]. For nonlinear scoring methods, SQI values are compressed into a narrower range, and regression statistics should be interpreted as descriptive of clustered agreement rather than indicative of predictive performance across the full SQI domain.

**Table 6. Sensitivity analysis of soil quality indices with across all the datasets, scoring and indexing methods.**

| Indexing method | Dataset | Scoring method | |
|---|---|---|---|
| | | Linear | Non-linear |
| | | Sensitivity | |
| SQIa-NT | TDS | 1.19 | 1.03 |
| | MDS_Corr | 1.16 | 1.03 |
| | MDS_PCA | 1.12 | 1.01 |
| SQIa-T | TDS | 1.20 | 1.03 |
| | MDS_Corr | 1.25 | 1.04 |
| | MDS_PCA | 1.08 | 1.04 |
| SQIw-NT | TDS | 1.05 | 1.04 |
| | MDS_Corr | 1.13 | 1.03 |
| | MDS_PCA | 1.12 | 1.01 |
| SQIw-T | TDS | 1.05 | 1.04 |
| | MDS_Corr | 1.13 | 1.03 |
| | MDS_PCA | 1.12 | 1.01 |

TDS: Total dataset; MDS_Corr: Minimum dataset selection based on correlation with corn yield; MDS_PCA: Minimum dataset selection based on principal component analysis; SQIa-NT: Soil quality index (additive) where no threshold values was used for data normalization; SQIa-T: Soil quality index (additive) where threshold values was used for data normalization; SQIw-NT: Soil quality index (additive weighted) where no threshold values was used for data normalization; SQIw-T: Soil quality index (additive weighted) where threshold values was used for data normalization.

Achieving a balance between universality and site specificity remains a central challenge in SQ framework development, as highlighted in global soil health syntheses [43,47]. Across all cases, the sensitivity of linear scoring methods was higher than that of nonlinear methods, clearly suggesting that linear methods are more suitable for SQI assessment. Similarly, linear scoring methods were found to be preferable after evaluating various linear and nonlinear approaches [17], whereas nonlinear methods were favored in other studies [16,48]. This discrepancy stems from earlier studies evaluating only single linear and nonlinear methods without incorporating sensitivity analysis.

Within linear scoring approaches, the sensitivity of the MDS evaluated using the SQIa was higher than that of the SQIw, regardless of whether threshold limits were applied. Notably, sensitivity for MDS$_{Corr}$ was significantly higher when thresholds were used (1.25) compared to when thresholds were not applied (1.16). In contrast, the SQIw method exhibited no significant sensitivity change with or without threshold application.

Overall, sensitivity analyses using Indiana site data as a reference provided critical insights into optimal scoring, indexing, and dataset selection strategies for SQI calculations. Based on these results, linear scoring methods with threshold limits, specifically MDS$_{Corr}$ and SQIa, were determined to be the most suitable. Accordingly, the LSM1 method (with thresholds) was selected as the most appropriate approach for SQI assessment because it consistently produced the highest sensitivity values, indicating greater responsiveness and differentiation among SQ indicators, while effectively leveraging the reduced MDS$_{Corr}$ dataset and the reliable performance of the SQIa indexing method.

### Evaluation of MDS$_{Corr}$, LSM1 (threshold), and additive soil quality indices

To validate the findings from the sensitivity analysis, SQIs were calculated using the developed MDS$_{Corr}$, LSM1, and SQIa across all study sites such as Indiana, Hoytville, Alabama, and Piketon. The degree of agreement between SQIs derived from MDS$_{Corr}$ and those from the TDS was evaluated using the Nash–Sutcliffe efficiency coefficient ($E_f$) and the relative deviation coefficient ($E_R$). The $E_f$ and $E_R$ values for Indiana, Hoytville, Alabama, and Piketon (Table 7). For MDS$_{Corr}$ based SQIs, $E_f$ values ranged from 0.65 to 0.94, exceeding commonly accepted performance thresholds ($E_f \geq 0.65$) and indicating good to excellent agreement with TDS across all locations. In contrast, SQIs calculated using MDS$_{PCA}$ yielded low or negative $E_f$ values, reflecting poor predictive performance relative to TDS. Similarly, $E_R$ values for MDS$_{Corr}$ based SQIs were consistently low (0.002 to 0.02), whereas MDS$_{PCA}$ based SQIs exhibited higher $E_R$ values (0.019 to 0.11), indicating greater deviation from the TDS benchmark. Because $E_f$ values closer to 1 and $E_R$ values closer to 0 represent stronger agreement and higher accuracy [39], the results in Table 7 demonstrate that SQI evaluations based on MDS$_{Corr}$ combined with LSM1 and additive indexing (SQIa) provide a robust and reliable approximation of TDS-derived SQIs. Overall, these

**Table 7. The Nash Effective coefficient ($E_f$) and relative deviation coefficient ($E_R$) for soil quality indices calclauted for Indiana, Alabama, and Hoytville and Piketon (Ohio) sites.**

| Location | Dataset | $E_f$ | $E_R$ |
|---|---|---|---|
| Indiana | TDS and MDS$_{Corr}$ | 0.94 | 0.002 |
| | TDS and MDS$_{PCA}$ | −1.57 | 0.110 |
| Hoytville | TDS and MDS$_{Corr}$ | 0.82 | 0.003 |
| | TDS and MDS$_{PCA}$ | 0.10 | 0.090 |
| Alabama | TDS and MDS$_{Corr}$ | 0.65 | 0.020 |
| | TDS and MDS$_{PCA}$ | −1.08 | 0.040 |
| Piketon | TDS and MDS$_{Corr}$ | 0.84 | 0.020 |
| | TDS and MDS$_{PCA}$ | −0.47 | 0.019 |

TDS: Total dataset; MDS$_{Corr}$: Minimum dataset selection based on correlation with corn yield and MDS$_{PCA}$: Minimum dataset selection based on principal component analysis.

findings confirm that $MDS_{Corr}$ represents a more accurate and transferable substitute for TDS than $MDS_{PCA}$ for SQI evaluation. closer to 0 indicate smaller deviations and higher accuracy of the SQI calculations based on the MDS [39].

## Logical interpretation of dataset selection and scoring of soil quality indexing

Each dataset, scoring approach, and indexing method influenced the SQI results for Indiana, Hoytville (Ohio), Alabama, and Piketon (Ohio) (Fig 7). The variability of SQI across different datasets, scoring methods, and indexing strategies was consistent with previous studies [1,8,15–17] confirming that these factors significantly contribute to SQI outcome variations.

The $MDS_{PCA}$ and $MDS_{Corr}$ in this study were developed using statistical techniques designed to reduce and eliminate redundant SQ indicators necessary for calculating the SQI. Notably, $MDS_{Corr}$ was established based on the direct correlation between potential SQ indicators and corn production over a five-year period under control treatment across four geographic locations. As agricultural production, even under standardized control treatments (no gypsum, no cover crop), inherently reflect the integrated effects of multiple SQ parameters, using crop yield as a deductive indicator reinforces the relevance and robustness of the $MDS_{Corr}$ selection method. The selected indicators for $MDS_{Corr}$ proved to be universal rather than site-specific, as confirmed by the results of sensitivity analysis, including sensitivity analysis results (Table 6) and model performance metrics (Table 7). This suggests that $MDS_{Corr}$ may be appropriate for assessing soil quality across a diverse set of soils. However, before a firm conclusion can be reached, the results need to apply across soils and climates.

Conversely, the $MDS_{PCA}$ method did not yield a consistent set of indicators across all sites, as each location required a unique combination of variables (S6–S9 Tables). This site-specific variability limits the broader applicability of $MDS_{PCA}$

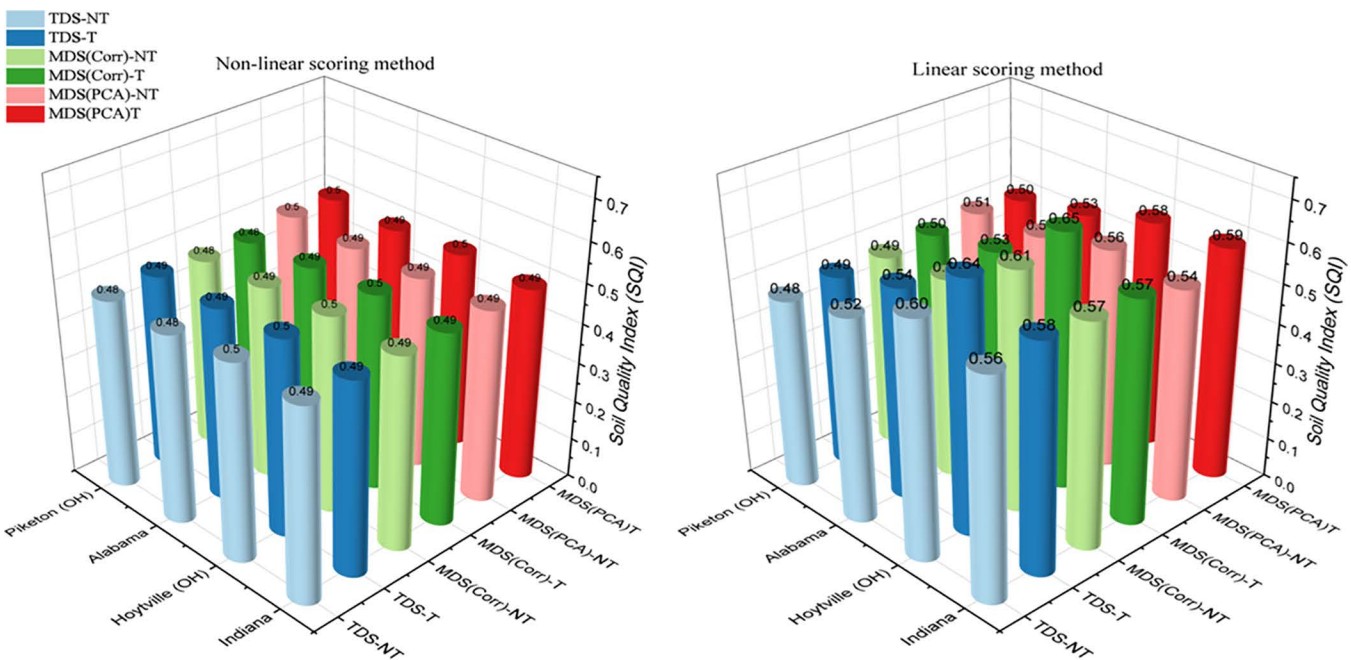

**Fig 7. Comparison of calculated soil quality indices using various datasets for all experimental sites.** (TDS-NT: Total dataset with no threshold values used; TDS-T: Total dataset with threshold values used; $MDS_{Corr}$-NT: Minimum dataset based on correlation with crop yield without any threshold values used; $MDS_{Corr}$-T: Minimum dataset based on correlation with crop yield with threshold values used; $MDS_{PCA}$-T: Minimum dataset based on principal component analysis with threshold values used; and $MDS_{PCA}$-NT: Minimum dataset based on principal component analysis without any threshold values used).

for developing a universal SQI framework. In contrast, the $MDS_{Corr}$ method, which selects indicators based on their direct correlation with crop yield, demonstrated greater consistency and practicality for multi-site assessments. Fig 7 further supports this conclusion, where SQI values derived from $MDS_{Corr}$ particularly under linear scoring with threshold application were consistently higher across all locations compared to TDS and $MDS_{PCA}$. These results highlight the robustness of the $MDS_{Corr}$ approach for dependable and interpretable SQI calculations across diverse agroecosystems.

Despite the widespread use of various normalization methods in SQI integration globally, no single method has achieved universal acceptance [12,16,17]. However, our results show that among the evaluated methods, $MDS_{Corr}$ combined with a linear scoring method demonstrated superior performance for SQI calculation. The linear scoring approach, as previously stated, consistently produced higher and more accurate SQI values across interstate sites, encompassing varied soil types and climatic conditions, compared to non-linear scoring approaches. Linear scoring maintains a direct, proportional relationship between the SQ indicators and the computed SQIs, avoiding the widespread variability and interpretative complications introduced by non-linear transformations. In contrast, non-linear scoring methods, particularly when applied to $MDS_{PCA}$ and other approaches, led to lower and more variable SQI scores due to the complex interactions among SQ indicators introduced by the transformations. Moreover, sensitivity, Nash efficiency ($E_f$), and relative deviation ($E_R$) analyses indicated that linear scoring methods better explain the observed variations than non-linear methods.

The distinction between $MDS_{Corr}$ with threshold (T) and non-threshold (NT) methods becomes evident when examining the SQI results across sites. The $MDS_{Corr}$ method, applying a threshold during data normalization, produced higher and more consistent SQI values, typically ranging from 0.57 to 0.59 for Indiana site with the linear scoring method (Fig 7). This indicates that threshold application stabilizes the SQI by minimizing the disproportionate influence of extreme or outlier values, leading to a more uniform SQ assessment across diverse sites like Piketon (Southern Ohio), Alabama, and Indiana. The threshold acts to moderate data variability, improving the reliability of SQI estimates, especially in areas with heterogeneous soil conditions.

Threshold-based normalization has previously been shown to reduce the influence of extreme values and improve comparability of SQI estimates across heterogeneous soils [15,21]. In contrast, $MDS_{Corr}$, without threshold application, resulted in slightly lower and more variable SQI values, ranging from 0.54 to 0.57 for Indiana with the linear scoring methods (Fig 7). This suggests that applying thresholds helps standardize the influence of individual indicators, reducing the impact of extreme or outlier values. As a result, SQ assessments became more uniform and comparable across diverse sites such as Piketon (Southern Ohio), Alabama, Hoytville (Northern Ohio), and Indiana. The threshold mechanism effectively moderates data variability, enhancing the reliability and interpretability of SQI estimates in heterogeneous soil environments. Thus, Fig 7 illustrates that $MDS_{Corr}$ with a threshold value is preferable when consistency and robustness across sites are critical, as it mitigates the impact of outliers and yields a more moderated SQ assessment. $MDS_{Corr}$ may also be more appropriate in situations where capturing the full natural variability of soil conditions is prioritized, albeit at the cost of increased result variability.

### Role of key soil properties controlling soil quality in the studied cropping systems

Beyond methodological comparisons, the correlation-based minimum dataset ($MDS_{Corr}$) provides insight into the soil properties that most strongly govern soil quality in the studied corn-based cropping systems. The dominance of SOC and TN reflects their significant role in nutrient cycling, biological activity, and overall soil functional capacity, consistent with established SQ frameworks [1,4,6,42,43]. The inclusion of AC further highlights the importance of labile carbon pools that respond rapidly to management and serve as early indicators of SQ change [2,14,29]. Physical indicators such as ρb and MaAS, MiAS, MWD, and GMD emphasize the critical role of soil structure in regulating aeration, water movement, root growth, SOC, and TN protection. Negative associations of ρb with SQI indicate constraints imposed by compaction, while aggregate stability metrics capture soil resilience against physical disturbance and carbon stabilization processes [6,20,36,49]. Together, these indicators suggest that soil quality in the evaluated systems is controlled by the interaction

between SOC, TN, AC, and structural integrity (ρb and aggregation metrics), rather than by single properties alone. This reinforces the relevance of MDS$_{Corr}$ as a process-based and agronomically meaningful SQ assessment approach [3,8,43].

## Feasibility of developing core indicator-based soil quality dataset

Previously, a MDS$_{Corr}$ was developed based on eight SQ indicators correlated with corn production: TN, SOC, AC, ρb, MaAS, MiAS, MWD, and GMD. When these indicators were correlated with corn grain yield (used as a deductive measure of SQ), several core indicators emerged as potential predictors of both SQ and crop yield. However, due to confounding relationships and multicollinearity among some of these indicators, their individual predictive reliability was limited. This underscored the need to further refine the indicator set by removing redundant variables and validating the most consistent predictors across diverse soil and management conditions.

To address issues of confounding effects and autocorrelation bias in multivariate SQ datasets, a refined set of core indicators was selected based on orthogonality, sensitivity to management practices, ease of routine measurement, consistency across sites, and contribution to explaining soil quality variability [5,12]. Two key indicators such as TN and MaAS were chosen to form a core minimum data set (CMDS) for SQ assessment. TN was selected as a composite chemical indicator due to its stoichiometric linkage with SOC in SOM through a typical C:N ratio of approximately 10:1 and its multifaceted role in regulating agroecosystem functions [12,50]. TN is associated with crop productivity, food quality, microbial activity and biodiversity, nutrient-use efficiency, enzymatic activity, SOC sequestration, and aggregate formation. However, we acknowledge concerns that TN and SOC are often highly correlated. While SOC is a more direct measure of SOM and plays a critical role in carbon cycling and structural stability, TN encompasses additional nitrogen-related dynamics important for soil fertility and microbial processes. Thus, TN is retained in CMDS not as a substitute for SOC but as a complementary indicator, offering a broader representation of soil biochemical health, especially in systems where nitrogen dynamics are tightly managed or variable. MaAS, selected as a composite physical indicator, reflects critical soil functions related to habitat suitability for microbes, SOC, and TN protection as particulate organic matter (POM), nutrient retention, soil porosity, water infiltration, and root penetration. Its sensitivity to tillage, cover cropping, and organic amendments makes it an early and effective indicator of physical soil health.

The results, presented in Fig 8, offer a comparative analysis of SQI values across different datasets such as TDS, MDS$_{Corr}$, MDS$_{PCA}$, and CMDS, using box plots for sites in Indiana, Hoytville, Alabama, and Piketon. The comparative analysis reveals important insights into the performance and feasibility of the CMDS approach.

At the Indiana, Hoytville, and Alabama sites, the SQI calculated using CMDS showed a wider distribution, and a higher range of values compared to TDS, MDS$_{Corr}$, and MDS$_{PCA}$. For instance, in Indiana, the SQI values for CMDS extended up to 0.73, significantly higher than MDS$_{Corr}$, which had a median SQI of 0.57 and a narrower interquartile range. Similarly, in Hoytville soils, CMDS displayed a higher median SQI (0.68) compared to MDS$_{Corr}$ (0.64). These results suggest that while CMDS captures more nuanced variations in SQ, the broader range could also indicate a need to incorporate additional indicators, particularly biological parameters, to refine the model and reduce variability. In Alabama soils, the SQI derived from CMDS also exhibited a broad distribution, ranging from 0.35 to 0.68. Despite the larger spread, CMDS captured high SQI values comparable to TDS and MDS$_{Corr}$, with a median SQI of 0.5. In contrast, MDS$_{Corr}$ produced more tightly clustered SQI values around 0.49, suggesting that its broader set of correlated parameters provides more consistent SQ assessments.

At Piketon (Southern Ohio) site, SQI results were more comparable across all methods. However, MDS$_{Corr}$ maintained a median SQI like other methods (approximately 0.5) with a smaller interquartile range, demonstrating its efficiency in matching TDS performance while using fewer parameters. CMDS again showed greater variability (ranging from 0.41 to 0.65), reflecting the impact of assessing SQ based on fewer indicators.

Overall, the results in Fig 8 suggest that CMDS, while promising in capturing critical soil functions, exhibits greater variability in SQI outcomes due to the exclusion of biological indicators and reliance on only two core parameters. In contrast,

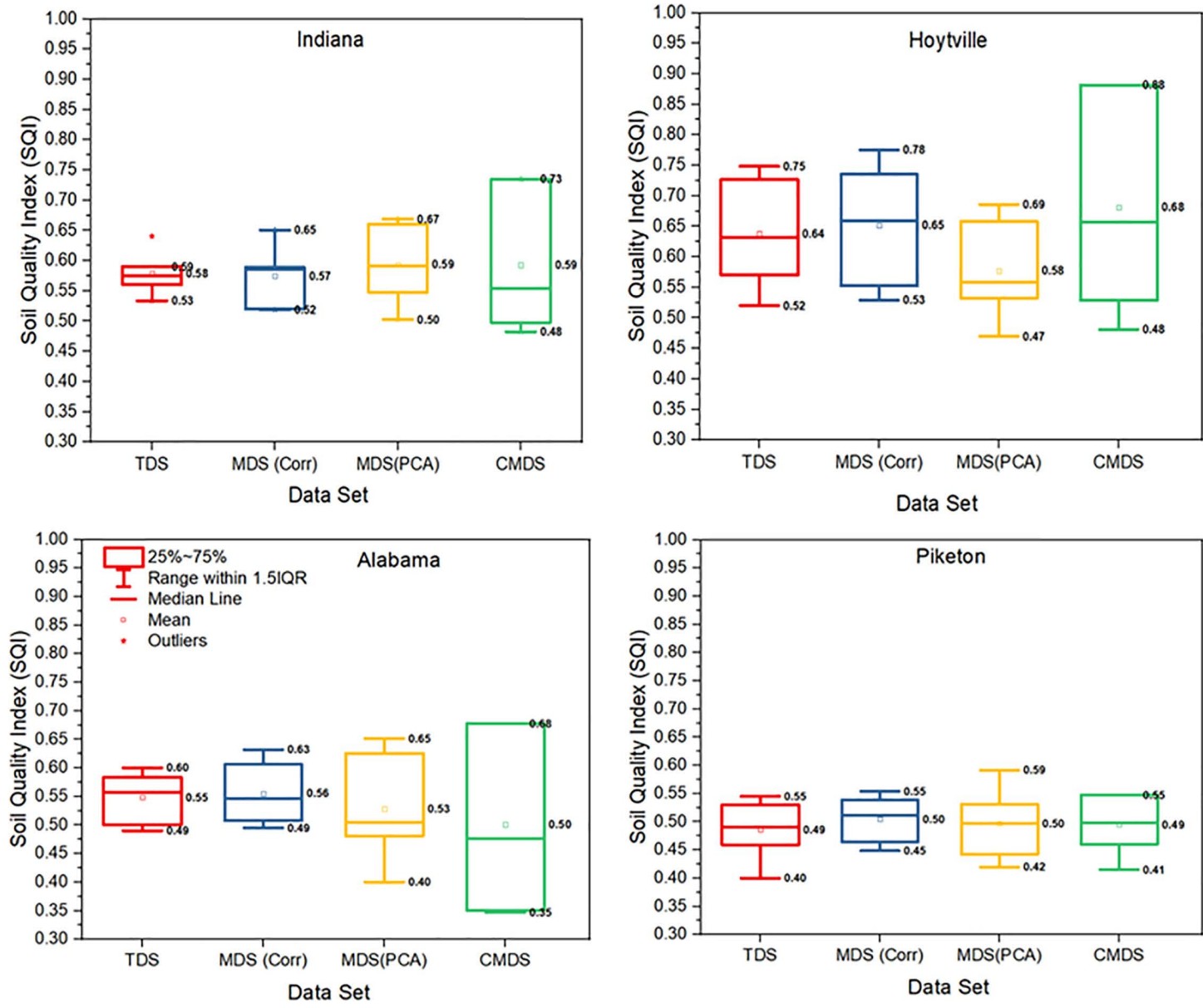

**Fig 8. Comparison of box plots for soil quality index (SQI) using three linear scoring (with threshold limit values), indexing methods [SQIa (additive), and SQIw (weghted additive).** [TDS: Total dataset; $MDS_{Corr}$: Minimum dataset selection based on correlation with crop yield; $MDS_{PCA}$: Minimum dataset selection based on principal component analysis; and CMDS: Critical minimum dataset].

$MDS_{Corr}$ consistently maintains a strong correlation with TDS across all sites, demonstrating its reliability and effectiveness as a more compact and practical method for SQ assessment. The close alignment of mean SQI values between $MDS_{Corr}$ and TDS, with differences within ±1%, further supports $MDS_{Corr}$ as the optimal choice for universal SQ assessment.

Future refinements of CMDS, particularly by incorporating key biological indicators, may enhance its accuracy and stability, making it a more balanced and robust tool for SQ assessments.

## Comparison of soil quality indices among sites

The comparison of SQI values among Indiana, Hoytville (Northern Ohio), Alabama, and Piketon (Southern Ohio) sites using TDS, $MDS_{PCA}$, and $MDS_{Corr}$ datasets, based on the LSM1 linear scoring method with and without threshold values and the SQIa indexing method, demonstrated that the linear scoring methods produced different SQI values for TDS, $MDS_{PCA}$, and $MDS_{Corr}$ (Fig 9). However, the general trend remained consistent regardless of the application of threshold limits, suggesting the robustness of the scoring methods.

When thresholds were applied (particularly for $MDS_{Corr}$, the SQI values were slightly higher, indicating that normalizing data within specific limits enhances the accuracy and stability of SQI calculations. This effect was especially noticeable at the Indiana and Hoytville (Northern Ohio) locations, where threshold-based SQI values (0.56 and 0.65, respectively) were significantly higher compared to those calculated without thresholds. Conversely, at sites like Piketon (Southern Ohio), the difference between threshold and no-threshold values was minimal, suggesting that the benefit of applying thresholds may vary by region.

Notably, the threshold-based $MDS_{Corr}$ approach provided more dependable and consistent SQI results across all four locations: Indiana (0.56), Hoytville (0.65), Alabama (0.55), and Piketon (0.51). These results closely aligned with those from the TDS method, highlighting the capacity of $MDS_{Corr}$ to effectively reduce the number of indicators without sacrificing accuracy. The application of thresholds helps prevent extreme values or outliers from disproportionately affecting the SQI calculation, thus improving the robustness of the assessment across diverse regions.

Overall, these results indicate that the linear scoring method LSM1, particularly when combined with threshold limits and the SQIa indexing method, provides a reliable and robust approach for SQ assessment using $MDS_{Corr}$. Using the

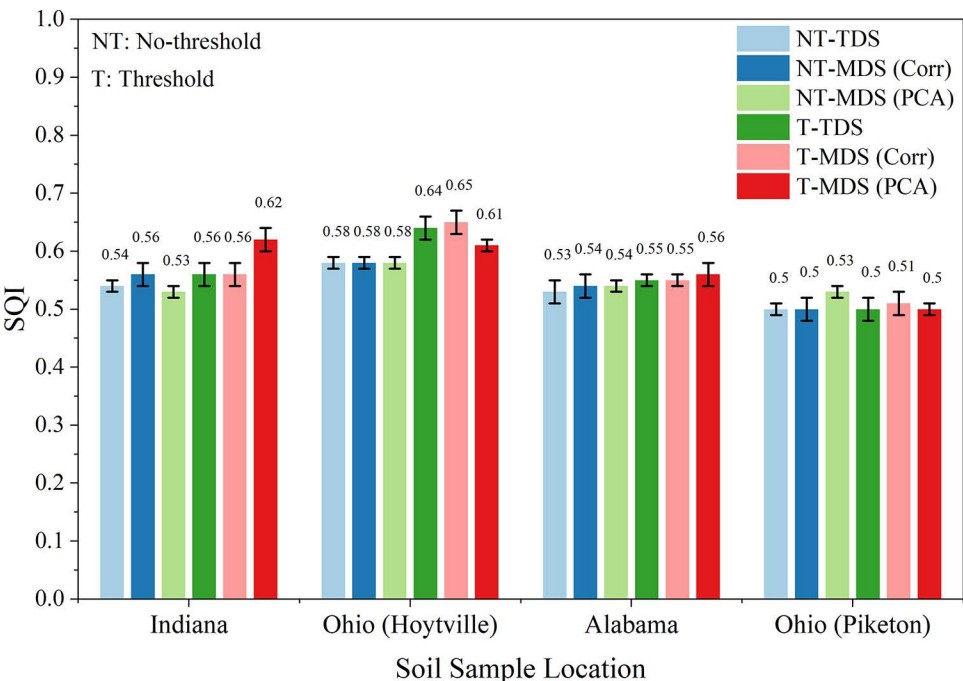

**Fig 9. Comparison of soil quality indices calculated using three linear scoring methods such as total dataset (TDS), minimum dataset based on correlation with crop yield ($MDS_{Corr}$) and minimum dataset based on principal component analysis ($MDS_{PCA}$).** The Indexing method used was SQIa. NT: No threshold value was used for data normalization; T: Threshold value was used for data normalization.

optimized methods, the average SQI values for Indiana, Hoytville, Alabama, and Piketon were 0.57±0.2, 0.61±0.1, 0.53±0.1, and 0.5±0.1, respectively.

## Conclusions

Soil quality indices are sensitive to indicator selection, scoring functions, and indexing methods, and no single approach is universally applicable across sites. Among the evaluated methods, the $MDS_{Corr}$ combined with LSM1 and $SQI_a$ consistently produced the most reliable, sensitive, and transferable SQI values across diverse soils and regions. The TDS approach exhibited high variability (~37%) in SQI values, due to inconsistencies in scoring and indexing methods. To enhance accuracy and consistency, scoring and indexing techniques were optimized using sensitivity analysis, the Nash efficiency coefficient ($E_f$), and the relative deviation coefficient (ER). We developed a $MDS_{Corr}$ comprising nine universally applicable soil indicators strongly associated with five years of corn productivity across the sites. SQI values derived from the $MDS_{Corr}$ showed better alignment with TDS when linear scoring was combined with additive or weighted additive indexing methods, compared to PCA-based indexing. Validation across all sites revealed that SQI differences between $MDS_{Corr}$ and TDS were within ±1%, supporting the use of MDSCorr as an effective and simplified alternate to TDS. We developed a $MDS_{Corr}$ comprising eight universally applicable soil indicators significantly associated with five years of corn productivity across the sites. SQI values derived from the $MDS_{Corr}$ showed better alignment with TDS when linear scoring was combined with additive or weighted additive indexing methods, compared to PCA-based indexing. Validation across all sites revealed that SQI differences between $MDS_{Corr}$ and TDS were within ±1%, supporting the use of $MDS_{Corr}$ as an effective and simplified alternate to TDS. Although the development of CMDS was explored, its higher variability in SQI distribution across sites limited its reliability relative to $MDS_{Corr}$. Based on these findings, we recommend using $MDS_{Corr}$ in conjunction with linear scoring and additive indexing for robust SQI calculation. The final SQI rankings across the sites were: Hoytville > Indiana > Alabama > Piketon. Future research should focus on validating the $MDS_{Corr}$ framework across a wider range of cropping systems and soil types, incorporating key biological indicators to further improve sensitivity to management-induced changes, and refining region-specific threshold values using long-term datasets. Integration of $MDS_{Corr}$ with digital soil mapping and decision-support tools would further enhance its applicability for site-specific soil management and monitoring.

## Supporting information

**S1 Table. Control treatment corn yield in Alabama, Indiana, and Ohio (Hoytville and Piketon) from 2012 to 2016 (average of four replications).**
(DOCX)

**S2 Table. Descriptive statistics of controlled treatment soil properties at Indiana site in 2012–2016 (average of 4 replications).**
(DOCX)

**S3 Table. Descriptive statistics of controlled treatment soil properties at Hoytville (Ohio) site (average of 4 replications).**
(DOCX)

**S4 Table. Descriptive statistics of controlled treatment soil properties of Alabama site (average of 4 replications).**
(DOCX)

**S5 Table. Descriptive statistics of controlled treatments soil properties at Piketon (Ohio) controlled soil (average of 4 replications).**
(DOCX)

**S6 Table. Load matrix and norm values of soil quality indicators evaluation for Indiana site (reference site).**
(DOCX)

**S6(a) Table. Pearson correlation coefficients among soil quality indicators and composite indices for the Indiana site (average of four replications, 2012–2016).**
(DOCX)

**S7 Table. Load matrix and norm values of soil quality indicators evaluation for Hoytville (Ohio) site.**
(DOCX)

**S7(a) Table. Pearson correlation coefficients among soil quality indicators and composite indices for the Hoytville (Ohio) site (average of four replications, 2012–2016).**
(DOCX)

**S8 Table. Load matrix and norm values of soil quality indicators evaluation for Alabama site.**
(DOCX)

**S8(a) Table. Pearson correlation coefficients among soil quality indicators and composite indices for the Alabama site (average of four replications, 2012–2016).**
(DOCX)

**S9 Table. Load matrix and norm values of soil quality indicators evaluation for Piketon (Ohio) site.**
(DOCX)

**S9(a) Table. Pearson correlation coefficients among soil quality indicators and composite indices for the Piketon (Ohio) site (average of four replications, 2012–2016).**
(DOCX)

**S10 Table. Soil quality threshold values for Hoytville site (Ohio).**
(DOCX)

**S11 Table. Soil quality threshold values for Alabama sites.**
(DOCX)

**S12 Table. Soil quality threshold values for Piketon (Ohio) sites.**
(DOCX)

**S13 Table. Calculation of the soil quality index using weighted additive method (SQIw) (LSM1-no threshold) and total dataset (TDS) for the Indiana site (reference site).** Calculated SQIw value was 0.59.
(DOCX)

**S14 Table. Calculation of the soil quality index using weighted additive method (SQIw) (LSM1-no threshold) and minimum dataset selection based on correlation with crop yield (MDS$_{Corr}$) for Indiana site.** The calculated SQIw value was 0.50.
(DOCX)

## Author contributions

**Funding acquisition:** Khandakar Islam, Norman Fausey, Tara VanToai.

**Resources:** Khandakar Islam, Javier Gonzalez, Dexter Watts, Norman Fausey, Randall Reeder, Dennis Flanagan.

**Writing – original draft:** Arifur Rahman.

**Writing – review & editing:** Khandakar Islam, Warren Dick, Vinayak Shedekar, Javier Gonzalez, Dexter Watts, Norman Fausey, Marvin Batte, Tara VanToai, Randall Reeder, Dennis Flanagan.

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
