## [Decision Letter · Decision Letter 0]

30 Dec 2025

Dear Dr. Islam,

Thank you for submitting your manuscript to PLOS ONE. After careful consideration, we feel that it has merit but does not fully meet PLOS ONE’s publication criteria as it currently stands. Therefore, we invite you to submit a revised version of the manuscript that addresses the points raised during the review process.

We look forward to receiving your revised manuscript.

Kind regards,

Roshan Babu Ojha

Academic Editor

PLOS One

Journal Requirements:

“United Soybean Board, Grant/Award Number: 1520-732-7226; The Ohio State University; USDA-ARS.”

“United Soybean Board, Grant/Award Number: 1520-732-7226; The Ohio State University; USDA-ARS.”

3. Please include captions for your Supporting Information files at the end of your manuscript, and update any in-text citations to match accordingly. Please see our Supporting Information guidelines for more information: http://journals.plos.org/plosone/s/supporting-information....

4. We note that Figure 9 in your submission contain [map/satellite] images which may be copyrighted. All PLOS content is published under the Creative Commons Attribution License (CC BY 4.0), which means that the manuscript, images, and Supporting Information files will be freely available online, and any third party is permitted to access, download, copy, distribute, and use these materials in any way, even commercially, with proper attribution. For these reasons, we cannot publish previously copyrighted maps or satellite images created using proprietary data, such as Google software (Google Maps, Street View, and Earth). For more information, see our copyright guidelines: http://journals.plos.org/plosone/s/licenses-and-copyright.

(1) You may seek permission from the original copyright holder of Figure 9 to publish the content specifically under the CC BY 4.0 license.

Reviewers' comments:

Reviewer's Responses to Questions

**Comments to the Author**

1. Is the manuscript technically sound, and do the data support the conclusions?

Reviewer #1: Yes

Reviewer #2: Yes

2. Has the statistical analysis been performed appropriately and rigorously?

Reviewer #1: Yes

Reviewer #2: No

3. Have the authors made all data underlying the findings in their manuscript fully available?

Reviewer #1: Yes

Reviewer #2: No

4. Is the manuscript presented in an intelligible fashion and written in standard English?

Reviewer #1: Yes

Reviewer #2: Yes

Reviewer #1: Reviews for

Minimum Dataset with Integrated Scoring and Indexing Methods for Soil Quality Assessment

Manuscript Number: PONE-D-25-51945

This submission contributes to SQ literature by integrating quantitative approaches. As contextual, SQ shows to be site specific. It is adequately reflective and compelling, and well written. Improvements as follows:

Inferring soil compaction from bulk density as noted by “increasing pb (compaction)” is likely unfound given there are contrasting regional sites where inherent attributes such as texture can be driving pb, instead of management. Soils with different pb can arrive and show the same detrimental compaction, and vice versa, different pb can result in the same yield outcome. For instance, sandy soils can have relatively high bulk density with exhibiting compaction as they are well aerated and pores are interconnected. As pb is not a functional variable, but mostly descriptive, its usefulness for the purpose of informing compaction in a SQ study is questionable or at least debatable due to introducing confounding influence.

Soil depth increments are presented in inconsistent way through the paper. Please cross check the document and improve this expression consistently.

For informing introduction and discussion sections, recent work on a wider range of cropping systems elsewhere has shown non-linear weighted additive indexing approach to outperform other options in terms of their sensitivity and effectiveness to arrive at SQ indexes – can this be reconciled in the text?. For example, as relevant for this aspect, refer and check: Iheshiulo et al. 2024. Quantitative evaluation of soil health based on a minimum dataset under various short-term crop rotations on the Canadian prairies. Science of The Total Environment 935, 173335

In particular, the discussion section shows lack of references to put the results into a broader context of the available literature.

“non-SMB” is puzzling in the submission. Although Table 3 indicates a statistically significant positive Pearson correlation with grain yield, Table 4 states “Less is better” for this variable and derivate. One would anticipate that having more carbon in non SMB pool (like in about any pool in soils) would result in improving ecosystem functions, including maize productivity and hence showing an upper asymptote while reaching saturation. In fact, non-SMB-C pretty much resembles SOC or TC data, which in Table 4 shows as “more is better”. Please elaborate, correct or address because this is concerning given that non-SMB was selected as part of MDS.

It feels disjointed and even redundant to read a paragraph starting as “In conclusion, the analysis” just before the section entitled "Conclusions”. Also initial sentences of the "Conclusions” section reads more as a narrative summary of the study while it would read better to directly state the key findings or lessons learned.

Right panel of Fig. 6 for non linear scoring method simply shows two clusters right-top corner, and left-bottom corner, and there are no data points in the intermediate range. From below 0.485 to above 0.495 in the horizontal x axis, this sizable open gap in the data makes unsubstantiate the provided robust linear regressions due to uncertainty. The presented R2, as high as 93%, are an artifact of having just two extreme clusters, drawing unsupported linear regressions in between, and nothing in between the clusters of data points to inform the goodness of fitness over the entire range, and enable definite statements in the text of the abstract and discussion section. Please address this aspect.

Reviewer #2: I went through the paper, and I found that generally, the topic is interesting. However, it needs major revisions before being accepted for publication. The comments are listed below:

• Abstract: It is better to present those properties that were recognized as crucial factors in the calculation of SQI in the studied region.

• Page 6 – Paragraph 1: Provide more details on the total number of soil data, and also the number of soil data studied in each site separately.

• Page 8 – Total dataset: First, the presentation of the properties is not appropriate and should be revised, particularly in terms of the use of parentheses and abbreviations to ensure clarity. Second, there are 11 chemical properties, not 12 — please correct this. In addition, specify that MWD and GMD are related to aggregate size distribution or primary particle size distribution.

• Page 8 – Selection of minimum dataset ...: Some of the properties selected for the MDS may be correlated with each other. For instance, MWD and GMD might show a strong and significant correlation. Therefore, including properties with correlation coefficients lower than 0.5 in relation to corn yield may lead to improved results. The Variance Inflation Factor (VIF) can be a useful indicator of multicollinearity among soil properties. It is recommended to select variables with correlation coefficients greater than ± 0.4 and retain only one of the two properties when their VIF exceeds 5. This approach could enhance the reliability of the analysis and potentially yield better results than the current method. At least, it would be worthwhile to test this procedure.

• Page 11: All equations should be numbered consecutively in the main text.

• Page 13 – The last paragraph: Blue line and red line of which figure? Address the related figure.

• Table 5: Define the abbreviations and symbols in the footnote of the table.

• Page 15 – The last paragraph: Correlation normally shows with r, not r2. Check this index. In addition, r² is always positive. Why was it calculated as r2 = - 0.52? Check and revise these statements.

• Page 16: correct (R2 = 0.27 0.7)

• Table 6: correct “sensityvity” in the title

• Page 18: Regarding the statement “The positive Ef values across all locations …”. Just being positive Ef values does not show a very good accuracy and superiority of a relationship. Ef value varies from - ∞ to 1. A proper relationship should have an Ef value more than 0.65 and not just positive. So, I suggest revising this sentence.

• Page 19: No need to define the Ef and ER again. Once defined in the main text is enough.

• Page 19: No need to mention “Supporting Tables S6-S9”. Just write as Tables S6-S9.

• Discussion section: Overall, the Discussion section requires improvement. The authors have primarily focused on the methods used for developing the SQI; however, it is also essential to discuss the key soil parameters identified and their relationships with soil quality in the studied field and cropping system. This is a critical aspect of the paper, and the Discussion section should be strengthened accordingly.

• Conclusion: Provide more specific suggestions for future works and studies.

Best Regards,

.

Reviewer #1: No

Reviewer #2: No

---

## [Author Response · Author response to Decision Letter 1]

2 Mar 2026

It is big file. Therefore, it has been attached.

---

## [Decision Letter · Decision Letter 1]

16 Mar 2026

Minimum Dataset with Integrated Scoring and Indexing Methods for Soil Quality Assessment

PONE-D-25-51945R1

Dear Dr. Islam,

We’re pleased to inform you that your manuscript has been judged scientifically suitable for publication and will be formally accepted for publication once it meets all outstanding technical requirements.

Kind regards,

Roshan Babu Ojha

Academic Editor

PLOS One

Additional Editor Comments (optional):

Reviewers' comments:

Reviewer's Responses to Questions

**Comments to the Author**

Reviewer #1: All comments have been addressed

2. Is the manuscript technically sound, and do the data support the conclusions?

Reviewer #1: Yes

3. Has the statistical analysis been performed appropriately and rigorously?

Reviewer #1: Yes

4. Have the authors made all data underlying the findings in their manuscript fully available?

Reviewer #1: Yes

5. Is the manuscript presented in an intelligible fashion and written in standard English?

Reviewer #1: Yes

Reviewer #1: (No Response)

.

Reviewer #1: No

---

## [Editor Report · Acceptance letter]

PONE-D-25-51945R1

PLOS One

Dear Dr. Islam,

I'm pleased to inform you that your manuscript has been deemed suitable for publication in PLOS One. Congratulations! Your manuscript is now being handed over to our production team.

Kind regards,

on behalf of

Dr. Roshan Babu Ojha

Academic Editor

PLOS One